# Clinical characteristics of 82 cases of death from COVID-19

**Bicheng Zhang**[1]☯, **Xiaoyang Zhou**[2]☯, **Yanru Qiu**[1]☯, **Yuxiao Song**[1]☯, **Fan Feng**[1], **Jia Feng**[1], **Qibin Song**[1]*, **Qingzhu Jia**[3]*, **Jun Wang**[4]*

1 Cancer Center, Renmin Hospital of Wuhan University, Wuhan, Hubei, China, 2 Cardiac Care Unit, Eastern Campus, Renmin Hospital of Wuhan University, Wuhan, Hubei, China, 3 Institute of Cancer, Xinqiao Hospital, Army Medical University, Chongqing, China, 4 Department of Oncology, The First Affiliated Hospital of Shandong First Medical University, Jinan, Shandong, China

☯ These authors contributed equally to this work.
* qibinsong@163.com (QS); qingzhu.jia@tmmu.edu.cn (QJ); ggjun2005@126.com (JW)

**Data Availability Statement:** All relevant data are within the manuscript and its Supporting Information files.

## Abstract

A recently developed pneumonia caused by SARS-CoV-2 bursting in Wuhan, China, has quickly spread across the world. We report the clinical characteristics of 82 cases of death from COVID-19 in a single center. Clinical data on 82 death cases laboratory-confirmed as SARS-CoV-2 infection were obtained from a Wuhan local hospital's electronic medical records according to previously designed standardized data collection forms. All patients were local residents of Wuhan, and a large proportion of them were diagnosed with severe illness when admitted. Due to the overwhelming of our system, a total of 14 patients (17.1%) were treated in the ICU, 83% of deaths never received Critical Care Support, only 40% had mechanical ventilation support despite 100% needing oxygen and the leading cause of death being pulmonary. Most of the patients who died were male (65.9%). More than half of the patients who died were older than 60 years (80.5%), and the median age was 72.5 years. The bulk of the patients who died had comorbidities (76.8%), including hypertension (56.1%), heart disease (20.7%), diabetes (18.3%), cerebrovascular disease (12.2%), and cancer (7.3%). Respiratory failure remained the leading cause of death (69.5%), followed by sepsis/MOF (28.0%), cardiac failure (14.6%), hemorrhage (6.1%), and renal failure (3.7%). Furthermore, respiratory, cardiac, hemorrhagic, hepatic, and renal damage were found in 100%, 89%, 80.5%, 78.0%, and 31.7% of patients, respectively. On admission, lymphopenia (89.2%), neutrophilia (74.3%), and thrombocytopenia (24.3%) were usually observed. Most patients had a high neutrophil-to-lymphocyte ratio of >5 (94.5%), high systemic immune-inflammation index of >500 (89.2%), and increased C-reactive protein (100%), lactate dehydrogenase (93.2%), and D-dimer (97.1%) levels. A high level of IL-6 (>10 pg/ml) was observed in all detected patients. The median time from initial symptoms to death was 15 days (IQR 11–20), and a significant association between aspartate aminotransferase (p = 0.002), alanine aminotransferase (p = 0.037) and time from initial symptoms to death was remarkably observed. Older males with comorbidities are more likely to develop severe disease and even die from SARS-CoV-2 infection. Respiratory failure is the main cause of COVID-19, but the virus itself and cytokine release syndrome-mediated damage to other organs, including cardiac, renal, hepatic, and hemorrhagic damage, should be taken seriously as well.

**Funding:** This research was funded by the National Nature Science Foundation of China (Number: 81272619 and 81572875), Natural Science Foundation of Hubei Province (2019CFB635), Health Commission of Hubei Province (WJ2019M194) and Fundamental Research Funds for the Central Universities, China. The funders had no role in study design, data collection and analysis, decision to publish, or preparation of the manuscript

**Competing interests:** The authors have declared that no competing interests exist.

## Introduction

In December 2019, the first acute respiratory disease case caused by severe acute respiratory syndrome coronavirus (SARS-CoV)-2, recently officially named Corona Virus Disease 2019 (COVID-19) by the World Health Organization (WHO), occurred in Wuhan, China [1, 2]. Person-to-person transmission has been identified through respiratory droplets or close contact [3–5]. By February 14, 2020, more than 60,000 confirmed cases and close to 2,000 cases of death have been documented in China, with hundreds of imported patients found in other countries [4–7].

Generally, the median incubation period of COVID-19 is 3 days (range: 0–24 days). Fever, cough, and fatigue are the most common symptoms [1]. Approximately 20–30% of cases develop severe illness, and some need further intervention in the intensive care unit (ICU) [8, 9]. Organ dysfunction, including acute respiratory distress syndrome (ARDS), shock, acute cardiac injury, and acute renal injury, can happen in severe cases with COVID-19 [1, 8, 9]. It has been reported that critically ill patients are more likely to be older, have underlying diseases, and have symptoms of dyspnea [9]. Oxygen therapy, mechanical ventilation, intravenous antibiotics and antiviral therapy are usually applied in clinical management, but presently, there are no effective drugs for improving the clinical outcome of COVID-19, especially for severe cases [1, 8, 9].

ARDS, a rapidly progressive disease, is the main cause of death for patients infected with previously recognized corona virus infections, such as SARS-CoV and Middle Eastern respiratory syndrome coronavirus (MERS-CoV) [10, 11]. In this context, it was initially considered that the lung is the most commonly damaged organ by SARS-CoV-2 infection since human airway epithelia express angiotensin converting enzyme 2 (ACE2), a host cell receptor for SARS-CoV-2 infection [12, 13]. However, increasing clinical cases have shown cardiac, renal and even digestive organ damage in patients with COVID-19 [9], which is consistent with the findings that kidney, colon and other tissues also express ACE2 in addition to airway epithelia [14, 15]. The above clinical phenomena and basic research suggest a more complicated pathogenesis of COVID-19. Hence, analyzing the clinical characteristics of death cases with COVID-19 is urgently needed to improve the outcome of infected patients.

## Methods

### Data collection

We retrospectively collected the epidemiological and clinical features of laboratory-confirmed COVID-19 patients who died from January 11, 2020, to February 10, 2020, in Renmin Hospital of Wuhan University. The confirmed diagnosis of COVID-19 was defined as a positive result by using real-time reverse-transcriptase polymerase chain reaction (RT-PCR) detection on nasopharyngeal swab. This study received approval from the Research Ethics Committee of the Renmin Hospital of Wuhan University, Wuhan, China (approval number: WDRY2020-K038). The Research Ethics Committee waived the requirement of informed consent before the study started because of the urgent need to collect epidemiological and clinical data. We analyzed all the data anonymously. The data set for this study is available in S1 Table.

The clinical features of the patients, including clinical symptoms, signs, laboratory analyses, radiological findings, treatment, and outcome, were obtained from the hospital's electronic medical records according to previously designed standardized data collection forms. Laboratory analyses included complete blood count, liver function, renal function, electrolyte test, coagulation function, C-reactive protein, lactate dehydrogenase, myocardial enzymes,

procalcitonin, and status of other viral infections. Radiological analyses comprised X-ray and computed tomography.

The date of symptom onset, initial diagnosis of COVID-19, and death were recorded accurately. The onset survival time was defined as the period between the onset of different symptoms and signs and the time of death. To increase the accuracy of the collected data, two researchers independently reviewed the data collection forms. Also, when the patients were admitted to hospital, the attending doctors directly communicated with them or their family members to ascertain epidemiological and symptom data.

## Inflammatory markers

Inflammatory markers associated with inflammation-linked disease were calculated using the specific parameters of blood tests. The neutrophil-to-lymphocyte ratio (NLR) was calculated by dividing the absolute neutrophil count by the lymphocyte count. The systemic immune-inflammation index (SII) was defined as platelet count × neutrophil count/lymphocyte count (/μL). Interleukin (IL)-6 was detected using the Human Cytokine Standard Assays panel (ET Healthcare, Inc., Shanghai, China) and the Bio-Plex 200 system (Bio-Rad, Hercules, CA, USA) according to the manufacturer's instructions.

## Statistical analysis

Descriptive analyses were used to determine the patients' epidemiological and clinical features. Continuous variables are presented as medians and interquartile ranges (IQRs), and categorical variables are expressed as percentages in different categories. The chi-squared test or Fisher's exact test was adopted for categorical variables. The associations between the different clinical variables and the time from initial symptoms to death were evaluated using Spearman's rank correlation coefficient. The statistical analyses in this study were performed using STATA 15.0 software (Stata Corporation, College Station, TX, USA). A two-sided p value less than 0.05 was considered statistically significant.

## Results

### The epidemiological features and underlying diseases

From January 11, 2020, to February 10, 2020, a total of 1,334 patients with a diagnosis of laboratory-confirmed COVID-19 were recorded at Renmin Hospital, Wuhan University, and 6.2% (82/1334) of these patients died. In the same period, the rates of mortality for all causes and non-COVID-19 in this hospital were 2.3% (162/7119) and 1.4% (80/5785), respectively. The mortality rate of COVID-19 was higher than that of non-COVID-19 ($p < 0.001$).

The epidemiological features and underlying diseases are shown in Table 1. All patients were local residents of Wuhan, and only 2 patients acknowledged contact with patients confirmed with SARS-CoV-2 infection. All the patients denied a history of contact with wildlife or the Huanan seafood market. A large proportion of the patients were diagnosed with severe illness when admitted (77/82). Severe illness meets any of the following: (a) Increased respiratory rate ($\geq$30 breaths/min), dyspnea, cyanosis of lips; (b) When breathing air, saturation of pulse oximetry (SPO) $\leq$93%; (c) Arterial partial pressure of oxygen (PaO2)/ concentration of oxygen inhalation (FiO2) $\leq$300mmHg (1mmHg = 0.133kPa); (d) Pulmonary imaging shows multiple lung lobe lesions or lesion progression within 48 hours >50%; (e) Combined with other clinical conditions requiring hospitalization. Critical illness meets any of the following: (a) Respiratory failure occurs and requires mechanical ventilation; (b) Shock occurs; (c) Combining other organ failure requires ICU monitoring treatment. Both severe and critical illness refer to

**Table 1. Clinical features of dead patients with COVID-19.**

| Clinical features | |
|---|---|
| **Age, years** | 72.5 (65.0–80.0) |
| **Age group** | |
| ≤40 years | 2/82 (2.4) |
| 40–50 years | 4/82 (4.9) |
| 50–60 years | 10/82 (12.2) |
| 60–70 years | 20/82 (24.4) |
| 70–80 years | 26/82 (31.7) |
| >80 years | 20/82 (24.4) |
| **Sex** | |
| Male | 54/82 (65.9) |
| Female | 28/82 (34.1) |
| **Local residents of Wuhan** | 82/82 (100) |
| **Severe pneumonia** | 77/82 (93.9) |
| **Comorbidity** | |
| Any | 62/82 (76.8) |
| Hypertension | 46/82 (56.1) |
| Diabetes | 15/82 (18.3) |
| Chronic obstructive pulmonary disease | 12/82 (14.6) |
| Heart disease | 17/82 (20.7) |
| Cerebrovascular | 10/82 (12.2) |
| Liver disease | 2/82 (2.4) |
| Renal insufficiency | 4/82 (4.9) |
| Infection | 5/82 (6.1) |
| Cancer | 6/82 (7.3) |
| Surgery | 3/82 (3.7) |
| Disease with reduced immunity | 14/82 (17.1) |
| **Number of comorbidity diseases** | |
| 0 | 19/82 (23.2) |
| 1 | 25/82 (30.5) |
| 2 | 18/82 (22.0) |
| ≥3 | 7/82 (8.5) |

Data are presented as median (IQR), n/N (%), where N represents the total number of patients with COVID-19

"Hubei province new coronavirus infection pneumonia diagnosis and treatment guidelines (trial first edition)". Most of the patients who died were male (65.9%), older than 60 years (80.5%), and the median age was 72.5 years (IQR 65.0–80.0). Most of the patients who died had comorbidities (75.6%), including hypertension (56.1%), heart disease (20.7%), diabetes (18.3%), cerebrovascular disease (12.2%), and cancer (7.3%). Thirty out of the 82 patients who died (30.5%) had 2 or more underlying diseases.

## Causes of death of patients with COVID-19

We analyzed the causes of mortality of the patients with laboratory-confirmed SARS-CoV-2 infection (Table 2). Respiratory failure remained the leading cause of death (69.5%), followed by sepsis/multiple organ dysfunction syndrome (MOF) (28.0%), cardiac failure (14.6%), hemorrhage (6.1%), and renal failure (3.7%). Furthermore, respiratory, cardiac, hemorrhagic, hepatic, and renal damage were found in 100%, 89%, 80.5%, 78.0%, and 31.7% of patients,

**Table 2. Causes of death of patients with COVID-19.**

| Causes of death and injured organs or systems | |
|---|---|
| **Causes of mortality** | |
| Respiratory | 57/82 (69.5) |
| Sepsis/MOF | 23/82 (28.0) |
| Cardiac | 12/82 (14.6) |
| Hemorrhage | 5/82 (6.1) |
| Renal | 3/82 (3.7) |
| Gastrointestinal | 2/82 (2.4) |
| Diabetic ketoacidosis | 2/82 (2.4) |
| Hepatic | 1/82 (1.2) |
| **Damaged organs or systems** | |
| Respiratory | 82/82 (100) |
| Cardiac | 73/82 (89.0) |
| Hemorrhage | 66/82 (80.5) |
| Hepatic | 64/82 (78.0) |
| Renal | 26/82 (31.7) |
| Gastrointestinal | 5/82 (6.1) |
| **Number of damaged organs or systems** | |
| 1 | 8/82 (9.8) |
| 2 | 3/82 (3.7) |
| 3 | 9/82 (11.0) |
| 4 | 41/82 (50.0) |
| ≥5 | 21/82 (25.6) |

Data are presented as n/N (%), where N represents the total number of patients with COVID-19.

respectively. A majority of patients (75.6%) had 3 or more damaged organs or systems following infection with SARS-CoV-2.

## Initial symptoms, laboratory analyses, radiological findings

As shown in Table 3, fever (78.0%), cough (64.6%), and shortness of breath (63.4%) were the main common symptoms. Diarrhea was observed in 12.2% of patients. All patients had bilateral involvement on chest radiographs. On admission, lymphopenia (89.2%), neutrophilia (74.3%), and thrombocytopenia (24.3%) were usually observed. All the patients had increased C-reactive protein levels (100%), and most patients had a high NLR of >5.0 (94.5%) and SII index of >500 (89.2%) as well as high lactate dehydrogenase (93.2%), D-dimer (97.1%), cardiac troponin I (86.7%), and procalcitonin (81.2%) levels. Insufficient cell immunity with reduced CD3+ (93.1%), CD8+ (98.3%), and CD6+CD56+ (100%) cell counts and high levels of circulating IL-6 (100%) were observed in the patients.

Furthermore, in the last 24 hours of death, lymphopenia (73.7%), neutrophilia (100%), and thrombocytopenia (63.2%) were continuously present. Increased C-reactive protein levels, high NLRs, and increased lactate dehydrogenase and increased D-dimer were found in all patients. The incidence of neutrophilia increased from 74.3% to 100%, and the incidence of lymphopenia was reduced from 89.2% to 73.7%. The incidence of high creatinine and blood urea nitrogen increased from 15.3% to 45.0% and 48.6% to 85.0%, respectively. High levels of IL-6 (>10 U/L) remained in all detected patients. More than half of the patients had a pH value of less than 7.35 (52.9%) and a $pO_2$ value of less than 60 mmHg (70.6%) (Table 4).

**Table 3. Initial symptoms and laboratory analyses of dead patients with COVID-19.**

| Initial clinical features, symptoms, laboratory analyses, treatment, and survival time | |
|---|---|
| **Initial symptoms** | |
| Fever | 64/82 (78.0) |
| Temperature, ˚C | 38.8 (38.0–39.0) |
| Fatigue | 38/82 (46.3) |
| Cough | 53/82 (64.6) |
| Nasal congestion | 1/82 (1.2) |
| Sore throat | 4/82 (4.9) |
| Diarrhea | 10/82 (12.2) |
| Vomiting | 2/82 (2.3) |
| Chest tightness | 36/82 (43.9) |
| Shortness of breath | 52/82 (63.4) |
| Consciousness problem | 17/82 (20.7) |
| **Complete blood count** | |
| Neutrophil count, $\times 10^9$/L | 6.8 (4.5–11.5) |
| Neutrophil count $<1.8 \times 10^9$/L | 2/74 (2.7) |
| Neutrophil count $>6.3 \times 10^9$/L | 55/74 (74.3) |
| Lymphocyte count, $\times 10^9$/L | 0.5 (0.3–0.8) |
| Lymphocyte count $<1.0 \times 10^9$/L | 66/74 (89.2) |
| Monocyte count, $\times 10^9$/L | 0.3 (0.2–0.5) |
| Platelet count, $\times 10^9$/L | 148.5 (102.0–206.0) |
| Platelet count $<100 \times 10^9$/L | 18/74 (24.3) |
| Platelet count $>400 \times 10^9$/L | 10/74 (13.5) |
| **Neutrophil-to-lymphocyte ratio** | 14.4 (7.1–25.8) |
| **Neutrophil-to-lymphocyte ratio $>5$** | 69/73 (94.5) |
| **Platelet-to-lymphocyte ratio** | 235.0 (259.0–442.0) |
| **Platelet-to-lymphocyte ratio $>200$** | 55/74 (74.3) |
| **Systemic immune-inflammation index** | 1966.1 (923.1–3206.5) |
| **Systemic immune-inflammation index $>500$** | 66/74 (89.2) |
| **Oxygen saturation, median (range), %** | 77.0 (65.5–85.0) |
| Oxygen saturation $<94\%$ | 27/28 (96.4) |
| **Blood biochemical analysis** | |
| C-reactive protein level, U/L | 11.7 (63.3–186.6) |
| C-reactive protein level $>10$U/L | 58/58 (100.0) |
| Alanine aminotransferase, U/L | 26.0 (18.5–47.5) |
| Alanine aminotransferase $>40$U/L | 22/72 (30.6) |
| Aspartate aminotransferase, U/L | 72.0 (30.0–71.0) |
| Aspartate aminotransferase $>40$U/L | 44/72 (61.1) |
| Total bilirubin, mmol/L | 13.6 (10.0–22.9) |
| Total bilirubin $>20.5$mmol/L | 22/72 (30.6) |
| Albumin, g/L | 33.1 (30.3–36.9) |
| Albumin $<40$g/L | 56/72 (77.8) |
| Potassium, mmol/L | 4.1 (3.7–4.4) |
| Potassium $>5.5$mmol/L | 16/72 (22.2) |
| Sodium, mmol/L | 141 (138.0–144.5) |
| Blood urea nitrogen, mmol/L | 8.6 (6.0–14.8) |
| Blood urea nitrogen $>8.8$mmol/L | 35/72 (48.6) |
| Creatinine, μmol/L | 78.0 (56.0–111.0) |
| Creatinine $>133$μmol/L | 11/72 (15.3) |

(*Continued*)

**Table 3.** (Continued)

| Initial clinical features, symptoms, laboratory analyses, treatment, and survival time | |
|---|---|
| Creatine kinase, U/L | 107.5 (56–336.5) |
| Creatine kinase >200 U/L | 25/72 (34.7) |
| Myoglobin, μg/L | 124.9 (71.1–392.3) |
| Myoglobin >110μg/L | 42/70 (60.0) |
| Lactate dehydrogenase, U/L | 515.0 (365.0–755.0) |
| Lactate dehydrogenase >250 U/L | 68/73 (93.2) |
| Creatine kinase-MB, ng/ml | 2.6 (1.2–5.3) |
| Creatine kinase-MB >5ng/ml | 21/70 (30.0) |
| NT-pro B-type natriuretic peptide, pg/ml | 122.0 (106.0–140.0) |
| Cardiac troponin T, pg/ml | 0.1 (0.1–0.7) |
| Cardiac troponin T >0.04pg/ml | 52/60 (86.7) |
| Procalcitonin, ng/ml | 0.3 (0.1–1.1) |
| Procalcitonin >0.1 ng/ml | 56/69 (81.2) |
| Prothrombin time, s | 13.2 (12.3–14.3) |
| Activated partial thromboplastin time, s | 29.4 (22.5–63.2) |
| D-dimer, mg/L | 5.1 (2.2–21.5) |
| D-dimer >0.55mg/L | 66/68 (97.1) |
| **Cell immunity, × 10$^9$/L** | |
| CD3+ cell count | 245.0 (45.6–67.8) |
| CD3+ cell count <723 × 10$^9$/L | 54/58 (93.1) |
| CD4+ cell count | 32.9 (26.0–42.1) |
| CD4+ cell count <404 × 10$^9$/L | 34/58 (58.6) |
| CD8+ cell count | 16.5 (10.9–26.5) |
| CD8+ cell count <220 × 10$^9$/L | 57/58 (98.3) |
| CD4+/CD8+ | 1.9 (1.2–3.0) |
| CD4+/CD8+>2 | 28/58 (48.3) |
| CD19+ cell count | 17.7 (10.3–25.5) |
| CD19+ cell count <80 × 10$^9$/L | 30/58 (51.7) |
| CD16+CD56+ cell count | 17.2 (11.6–27.5) |
| CD16+CD56+ cell count <84 × 10$^9$/L | 58/58 (100) |
| **Humoral immunity, g/L** | |
| IgG 7 | 12.9 (10.8–16.8) |
| IgG 7 >7g/L | 55/56 (98.2) |
| IgM | 0.9 (0.7–1.3) |
| IgM >0.4g/L | 52/56 (92.9) |
| IgA | 2.6 (1.9–3.7) |
| IgA >0.7g/L | 56/56 (100) |
| IgE | 61.5 (26.2–155.5) |
| IgE >100g/L | 23/56 (96.4) |
| C3 | 0.9 (0.8–1.1) |
| C3>0.9g/L | 35/56 (62.5) |
| C4 | 0.2 (0.2–0.3) |
| C4>0.1g/L | 54/56 (96.4) |
| **Interleukin 6, pg/ml** | 93.8 (64.5–258.0) |
| **Interleukin 6 >10pg/ml** | 11/11 (100) |
| **Nosocomial infection** | 5/82 (6.1) |
| **Intensive care unit admission** | 14/82 (17.1) |

Data are presented as median (IQR), or n/N (%), where N represents the total number of patients with COVID-19 with available data.

**Table 4. Laboratory analyses of dead patients in the last 24 hours of the death.**

| Laboratory analyses | |
|---|---|
| **Complete blood count** | |
| Neutrophil count, $\times 10^9$/L | 12.9 (11.3–26.0) |
| Neutrophil count >6.3 $\times 10^9$/L | 19/19 (100) |
| Lymphocyte count, $\times 10^9$/L | 0.5 (0.4–1.3) |
| Lymphocyte count <1.0 $\times 10^9$/L | 14/19 (73.7) |
| Monocyte count, $\times 10^9$/L | 0.4 (0.3–0.8) |
| Platelet count, $\times 10^9$/L | 77 (62.0–147.0) |
| Platelet count <100 $\times 10^9$/L | 12/19 (63.2) |
| Neutrophil-to-lymphocyte ratio | 19.6 (17.0–33.0) |
| Neutrophil-to-lymphocyte ratio >5 | 19/19 (100) |
| Platelet-to-lymphocyte ratio | 77.0 (62.0–147.0) |
| Platelet-to-lymphocyte ratio >200 | 7/19 (36.8) |
| Systemic immune-inflammation index | 1861.8 (950.8–3881.8) |
| Systemic immune-inflammation index >500 | 74/19 (100) |
| **Blood biochemical analysis** | |
| C-reactive protein level, U/L | 84.9 (73.5–186.6) |
| C-reactive protein level >10U/L | 13/13 (100) |
| Alanine aminotransferase, U/L | 30.5 (22.0–102.5) |
| Alanine aminotransferase >40U/L | 8/20 (40.0) |
| Aspartate aminotransferase, U/L | 74.5 (35.5–184.0) |
| Aspartate aminotransferase >40U/L | 14/20 (70.0) |
| Albumin, g/L | 31 (28.9–32.6) |
| Albumin <40g/L | 18/20 (90.0) |
| Potassium, mmol/L | 4.3 (3.9–4.7) |
| Sodium, mmol/L | 147.0 (143.0–156.0) |
| Blood urea nitrogen, mmol/L | 18.4 (8.5–33.5) |
| Blood urea nitrogen >8.8mmol/L | 17/20 (85.0) |
| Creatinine, μmol/L | 123 (87.5–361.5) |
| Creatinine >133μmol/L | 9/20 (45.0) |
| Creatine kinase, U/L | 258.0 (135.0–535.0) |
| Creatine kinase >200 U/L | 15/21 (71.4) |
| Myoglobin, μg/L | 342.5 (136.2–1000.0) |
| Myoglobin >110μg/L | 13/15 (86.7) |
| Lactate dehydrogenase, U/L | 784 (484.0–1,216.0) |
| Lactate dehydrogenase >250 U/L | 19/19 (100) |
| Creatine kinase-MB, ng/ml | 4.1 (2.7–7.0) |
| Creatine kinase-MB >5ng/ml | 5/15 (33.3) |
| NT-pro B-type natriuretic peptide, pg/ml | 3046.5 (1423.0–14262.0) |
| NT-pro B-type natriuretic peptide >1800 pg/ml | 10/14 (71.4) |
| Cardiac troponin T, pg/ml | 0.7 (0.1–8.7) |
| Cardiac troponin T >0.04 | 10/15 (66.7) |
| Procalcitonin, ng/ml | 1.1 (0.4–6.4) |
| Procalcitonin >0.1 ng/ml | 14/16 (87.5) |
| Prothrombin time, s | 17.2 (14.3–26.8) |
| Prothrombin time >12.1s | 13/13 (100) |
| Activated partial thromboplastin time, s | 32.8 (31.3–37.2) |
| D-dimer, mg/L | 42.7 (9.9–74.0) |

(*Continued*)

**Table 4.** (Continued)

| Laboratory analyses | |
|---|---|
| D-dimer >0.55mg/L | 13/13 (100) |
| **Interleukin-6, pg/ml** | 258.0 (105.6–262.4) |
| **Interleukin-6 >10pg/ml** | 11 (100) |
| Arterial gas | |
| PH | 7.3 (7.0–7.4) |
| PH <7.35 | 9/17 (52.9%) |
| PH >7.45 | 4/17 (23.5%) |
| $pO_2$ | 47 (40–61) |
| $pO_2$ <60 mmHg | 12/17 (70.6) |
| $pCO_2$ | 32.0 (28.0–42.0) |
| $pCO_2$ <35 mmHg | 9/17 (52.9) |
| $pCO_2$ >50 mmHg | 4/17 (23.5) |

Data are presented as median (IQR), n/N (%), where N represents the total number of patients with COVID-19 with available data

## Treatment and survival time of dead patients with COVID-19

As shown in S2 Table, a total of 14 patients (17.1%) were treated in the ICU. Patients received oxygen therapy (100%) and mechanical ventilation (40.2%), including 4 with invasive mechanical ventilation (4.8%). All patients received intravenous antibiotics and antiviral medications, and systematic corticosteroids were used in 29 patients (35.3%).

The median time from initial symptom onset to death was 15 days (IQR 15–20), and a significant association between aspartate aminotransferase (p = 0.002), alanine aminotransferase (p = 0.037) and time from initial symptoms to death was remarkably observed (Fig 1A–1C).

## Discussion

The mortality of 6.2% from the current study was lower than that of SARS infection in 2003. However, the mortality rate from this center is slightly higher than that previously reported [9]. We speculated that the reason might be that fewer patients in our study were transferred to the ICU in time when their condition rapidly worsened owing to the shortage of beds and medical supplies. On the other hand, limited death cases and fewer than 15 patients were

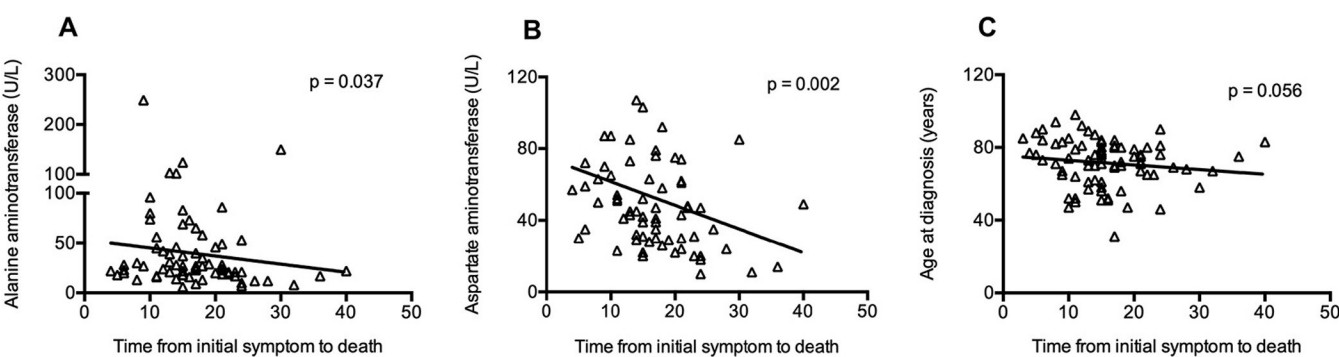

**Fig 1. The association between clinical features and time from initial symptoms to death.** (A) Alanine aminotransferase; (B) Aspartate aminotransferase; (C) Age.

included in their cohorts [8, 9], while many more death cases were included in the present study.

Our study first focused on the epidemiological characteristics of patients with COVID-19 who died and the mortality rate of COVID-19 was found higher than that of non-COVID-19 (p<0.001). Several factors were responsible for the death of the patients included in this study. A majority of the patients in our study were older than 60 years, and a borderline significant association between age and time from initial symptoms to death was significantly observed. These results are consistent with the finding that critically ill patients are more likely to be older [16]. Moreover, we found that underlying diseases such as hypertension, heart disease and diabetes were very common in our death cases, and 30.5% of patients had 2 or more comorbidities. These features are consistent with a previous report that patients with underlying diseases are more likely to develop severe illness [9]. Cancer patients comprised 7.3% of our cohort, which had a much higher morbidity than those without cancer, suggesting that cancer patients are more likely to develop severe disease or die. These results are consistent with the findings from a nationwide analysis in China [17]. Immune deficiency to virus infection seems to be a common feature in older males with comorbidities.

We further analyzed the cause of death of patients with COVID-19 and found that respiratory failure remained the leading cause of death. It has been reported that the binding receptor for SARS-CoV-2, ACE2, is mainly expressed in blood vessels and lung alveolar type II (AT2) epithelial cells [13]. Similar to SARS-CoV and MERS-CoV, SARS-CoV-2 can directly attack ACE2-expressing cells [13]. Indeed, pathological findings indicated that lungs infected with SARS-CoV-2 present with ARDS, pulmonary edema with hyaline membrane formation, and evident desquamation of pneumocytes [18]. Therefore, our finding that respiratory failure is the leading cause of death is consistent with the underlying pathological mechanism of COVID-19.

In addition to respiratory failure, cardiac failure, hemorrhage, renal failure and even MOF were also recognized as the cause of death by COVID-19 in our study. Laboratory findings also revealed cardiac, hepatic, and renal damage in some patients. In addition, we observed a significant association between aspartate aminotransferase, alanine aminotransferase and time from initial symptoms to death. These clinical phenomena could be explained by virus attack and cytokine release syndrome (CRS)-mediated tissue damage. First, ACE2 expression is also found in the kidney, heart, and liver; therefore, SARS-CoV-2 could invade the cells of the above tissues, reproduce and damage these organs [14, 15]. Second, viral infection and subsequent tissue damage either in the lung or other target organs could elicit immune cells to produce pro-inflammatory cytokines, namely, CRS, ultimately injuring the tissue and causing target organ failure. Obviously, increased amounts of cytokines, including IL-1β, IL-6, and monocyte chemotactic protein-1 (MCP-1), are associated with severe lung injury in patients infected with SARS-CoV and MERS-CoV [19, 20]. A recent study showed that high levels of IL-1β, interferon γ-induced protein 10, and MCP-1 occurred in the serum of patients infected with SARS-CoV-2, which probably led to the activation of the T-helper-1 cell response [1]. In the present report, high levels of IL-6 of more than 10 U/L and C-reactive protein were detected in all patients, even in the last 24 hours prior to death.

We also depicted the immune status of COVID-19 patients with severe illness. Most of the patients in our study presented with neutrophilia and lymphopenia on admission; specifically, reduced CD3+, CD4+, and CD8+ T-cell counts were observed in some patients. A high NLR was also observed on admission and 24 hours before death. These results, consistent with the previous findings found in patients with severe illness, suggest that the perturbation of the immune system contributes to the pathogenesis of SARS-CoV-2. These observations could also explain why older males with comorbidities likely succumb to COVID-19.

Our study has some limitations. First, some patients did not receive timely supportive interventions, such as admission to the ICU, because an increasing number of severe patients occurred in a short period and the overwhelming of our medical system. However, the present data could partially be a scenario where COVID-19 patients progress in a natural pathophysiological manner rather than outcomes from intervention by treatment. Second, the consecutive detection of cytokines is lacking, which fails to truly monitor the severity of CRS. Third, organ damage could originate from a history of medication, including nonsteroidal anti-inflammatory drugs, antibiotics, and traditional Chinese medicine, which are associated with renal or liver injury [21, 22]. In our study, all patients received intravenous antibiotics and antiviral drugs.

Overall, from the point of view of the causes of death, we presented the clinical characteristics of patients with COVID-19. Lung injury begins with an insult to the lung epithelium mainly attacked by SARS-CoV-2 itself because of ACE2 expressed in the lungs, which leads most commonly to respiratory failure. Other organs or tissues, more or less, are potentially damaged through direct attack from SARS-CoV-2. In addition, damage to multiple systems, including the lungs, might originate with systemic damage due to CRS following SARS-CoV-2 infection. Considering the pandemic potential and moderate threat of COVID-19 for populations with multiple underlying diseases, further studies are needed to focus on the pathology and pathophysiology of the tissue injury caused by SARS-CoV-2 infection, especially on the activation process of the immune response and cytokine storm.

## Supporting information

**S1 Table. Dataset.**
(XLSX)

**S2 Table. Treatment and survival time of dead patients with COVID-19.**
(DOCX)

## Acknowledgments

We acknowledge all health-care workers involved in the diagnosis and treatment of patients in Eastern Campus, Renmin Hospital of Wuhan University.

## Author Contributions

**Conceptualization:** Qibin Song, Qingzhu Jia, Jun Wang.

**Formal analysis:** Fan Feng, Jia Feng.

**Investigation:** Fan Feng, Jia Feng.

**Methodology:** Xiaoyang Zhou, Yanru Qiu.

**Writing – original draft:** Bicheng Zhang.

**Writing – review & editing:** Yuxiao Song, Qibin Song, Qingzhu Jia, Jun Wang.

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
