## [Decision Letter · Decision Letter 0]

9 Jun 2020

PONE-D-20-07999

Clinical characteristics of 82 cases of death from COVID-19

PLOS ONE

Dear Dr. wang,

Thank you for submitting your manuscript to PLOS ONE. After careful consideration, we feel that it has merit but does not fully meet PLOS ONE’s publication criteria as it currently stands. Therefore, we invite you to submit a revised version of the manuscript that addresses the points raised during the review process.

Please respond to all comments made by the reviewer and make changes to your manuscript accordingly. 

We look forward to receiving your revised manuscript.

Kind regards,

Xia Jin, MD, PhD

Academic Editor

PLOS ONE

Journal Requirements:

Reviewers' comments:

Reviewer's Responses to Questions

**Comments to the Author**

1. Is the manuscript technically sound, and do the data support the conclusions?

Reviewer #1: Yes

2. Has the statistical analysis been performed appropriately and rigorously? 

Reviewer #1: Yes

3. Have the authors made all data underlying the findings in their manuscript fully available?

Reviewer #1: Yes

4. Is the manuscript presented in an intelligible fashion and written in standard English?

Reviewer #1: Yes

5. Review Comments to the Author

Reviewer #1: This article is timely given the present pandemic for COVID 19> It is a descriptive retrospective review of the epidemiological descriptions of over 80 deaths at a single hospital in Wuhan in the first month of the epidemic. as such it is an informative article.

There are a few issues of note that need to be addressed:

1.) There is no definition of "severe Illness". Results section paragraph 2 line 4 77/82 were admitted with severe illness.

2,) Line 4 of paragraph 2 of Results. Incubation time is unclear how it can be calculated. The definition in method noted this calculation to be from time of contact of transmission exposure to date to symptoms. Only 2 patients reported exposure so is incubation time calculated for only those 2. If so, I don't find it very informative or helpful.

3.) Table 1 Comorbidity - first listing is "ALL" Does that mean ANY? Not clear what that means.

4.) Table 1 Comorbidity - please define disease of reduced immunity- is this Collagen vascular disease> is this HIV? what does this mean?

5.) Table 3 needs clarification and some editing....right now the list is just too long and difficult to find your way through--for example it notes Neutrophyl count ( is that a mean??) then lists % with very low and very high. This was the same for every value given. There must be an easier clearer way to present this data.

6.) Table 3 notes radiographic data in the table but it really lists only Bilateral infiltrates at 100% of patients. This would be easier to be removed form the table and listed at the time of the statement o "severe illness on presentation - is this in fact what you meant?

7.) Table 3 what is acquired infection? nosocomial infection during admission?

8.) The surprise is that only 17% of patients go to MICU --- this appears first in the table 3 and then later there is a brief mention of this in the discussion. This is important and needs to move up much sooner in your paper. 83% of deaths never received Critical Care Support due to the overwhelming of your system. Only 40% had mechanical ventilation support despite 100% needing oxygen and the leading cause of death being pulmonary. This all speaks to the overwhelming of the system and needs to be highlighted. You might look to see if the patients who did make it into the MICu were any different than the ones who did not?

9.) What anti viral medications were given? Table 3 again.

I think the paper is important glimpse into the early phase of the pandemic and is publishable with a bit of more work and editing.

6. PLOS authors have the option to publish the peer review history of their article (what does this mean?). If published, this will include your full peer review and any attached files.

Reviewer #1: Yes: E Jane Carter

---

## [Author Response · Author response to Decision Letter 0]

15 Jun 2020

PLOS ONE

Journal Requirements:

Response: The style has been checked and modified.

Response: There are no restrictions. The data set is uploaded as Supporting Information files (.xlsx).

Response: ORCID: 0000-0003-3941-2507 for Dr. Wang has been validated in Editorial Manager and added to the manuscript.

Response: Captions and files for Supporting Information has been included and updated, in-text citations are matched.

Response to reviewers

Reviewer #1: 

1. This article is timely given the present pandemic for COVID 19> It is a descriptive retrospective review of the epidemiological descriptions of over 80 deaths at a single hospital in Wuhan in the first month of the epidemic. as such it is an informative article. There are a few issues of note that need to be addressed:

There is no definition of "severe Illness". Results section paragraph 2 line 4 77/82 were admitted with severe illness.

Response: The definition has been added in the revised manuscript.

2. Line 4 of paragraph 2 of Results. Incubation time is unclear how it can be calculated. The definition in method noted this calculation to be from time of contact of transmission exposure to date to symptoms. Only 2 patients reported exposure so is incubation time calculated for only those. If so, I don't find it very informative or helpful.

Response: Yes, the situation at the time was too difficult to accurately trace and calculate the incubation period of each patient. Maybe, patients actually had exposure, but they did not acknowledge contact with patients confirmed with SARS-CoV-2 infection. So, we delete the data for the median incubation time (Page 11; Table 1).

3. Table 1 Comorbidity - first listing is "ALL" Does that mean ANY? Not clear what that means.

Response: “All” means patients having any one or more comorbidities. We have changed “All” to “Any” (Table 1).

4. Table 1 Comorbidity - please define disease of reduced immunity- is this Collagen vascular disease> is this HIV? what does this mean?

Response: It means abnormal blood examination for cellular immune or humoral immune indexes (Table 1).

5. Table 3 needs clarification and some editing...right now the list is just too long and difficult to find your way through--for example it notes Neutrophil count (is that a mean??) then lists % with very low and very high. This was the same for every value given. There must be an easier clearer way to present this data. 

Response: Neutrophil count is that a median (IQR). In fact, in all Tables data are presented as median (IQR), or n/N (%), where N represents the total number of patients with COVID-19 with available data. Please see the footnotes affiliated with table 3. To present clearly, “treatment” and “survival time” in table 3 have been separated to Supporting Information part as S2 Table at the end of the manuscript.

6. Table 3 notes radiographic data in the table, but it really lists only Bilateral infiltrates at 100% of patients. This would be easier to be removed from the table and listed at the time of the statement o "severe illness on presentation - is this in fact what you meant? 

Response: Bilateral infiltrates at 100% of patients are radiographic findings, did not equal severe illness on presentation. To present clearly, we delete this data in Table 3.

7. Table 3 what is acquired infection? nosocomial infection during admission? 

Response: Yes, it means nosocomial infection during admission. 

8. The surprise is that only 17% of patients go to MICU --- this appears first in the table 3 and then later there is a brief mention of this in the discussion. This is important and needs to move up much sooner in your paper. 83% of deaths never received Critical Care Support due to the overwhelming of your system. Only 40% had mechanical ventilation support despite 100% needing oxygen and the leading cause of death being pulmonary. This all speaks to the overwhelming of the system and needs to be highlighted. You might look to see if the patients who did make it into the MICU were any different than the ones who did not?

Response: The actual situation at the time was that the hospital was designated by Wuhan government as the COVID 19-designated hospital for severe cases, but the original 16 ICU beds were apparently not enough, medical staff and ventilators were extremely insufficient, and only 17% of patients go to MICU. Accordingly, there was a high mortality rate. After February 14th, as medical teams and supplies from all over the country went to and aid Wuhan, all the general wards in the hospital were transformed into ICUs and equipped with sufficient medical staff and equipment. After that, the mortality rate gradually declined. Thus, we highlighted the situation of the overwhelming of our medical system in Abstract.

9. What antiviral medications were given? Table 3 again.

Response: From the end of January to early February, few suitable antiviral drugs for severe patients are available. We tried LianHuaQingWen Capsule, Abidor, Ribavirin or Oseltamivir, but none seemed to have explicit effect.

---

## [Editor Report · Decision Letter 1]

17 Jun 2020

Clinical characteristics of 82 cases of death from COVID-19

PONE-D-20-07999R1

Dear Dr. wang,

We’re pleased to inform you that your manuscript has been judged scientifically suitable for publication and will be formally accepted for publication once it meets all outstanding technical requirements.

Kind regards,

Xia Jin, MD, PhD

Academic Editor

PLOS ONE
---

## [Editor Report · Acceptance letter]

29 Jun 2020

PONE-D-20-07999R1 

Clinical characteristics of 82 cases of death from COVID-19 

Dear Dr. Wang:

I'm pleased to inform you that your manuscript has been deemed suitable for publication in PLOS ONE. Congratulations! Your manuscript is now with our production department. 

Kind regards, 

on behalf of

Dr. Xia Jin 

Academic Editor

PLOS ONE